# Robust Self-Testing of Four-Qubit Symmetric States

**DOI:** 10.3390/e24071003

**Published:** 2022-07-20

**Authors:** Daipengwei Bao, Xiaoqing Tan, Qingshan Xu, Haozhen Wang, Rui Huang

**Affiliations:** School of Information Science and Technology, Jinan University, Guangzhou 510632, China; baodaipengwei@stu2020.jnu.edu.cn (D.B.); xuqingshan1008@stu2018.jnu.edu.cn (Q.X.); wanghz0125@163.com (H.W.); huang6666@foxmail.com (R.H.)

**Keywords:** Bell inequality, self-testing, symmetric states, device independent

## Abstract

Quantum verification has been highlighted as a significant challenge on the road to scalable technology, especially with the rapid development of quantum computing. To verify quantum states, self-testing is proposed as a device-independent concept, which is based only on the observed statistics. Previous studies focused on bipartite states and some multipartite states, including all symmetric states, but only in the case of three qubits. In this paper, we first give a criterion for the self-testing of a four-qubit symmetric state with a special structure and the robustness analysis based on vector norm inequalities. Then we generalize the idea to a family of parameterized four-qubit symmetric states through projections onto two subsystems.

## 1. Introduction

In recent years, quantum technology has developed rapidly and is expected to gain new real-world applications in communication, simulation, sensing, and computing [1,2,3,4]. Quantum devices promise to effectively solve some problems that are difficult to deal with in the classical field [5,6]. However, it also brings a thorny problem. How do we verify the solutions? The task of ensuring the correct operations of quantum devices in terms of accuracy of output is known as quantum verification [7], which is attracting more attention.

A common quantum state verification technology was quantum state tomography (QST) [8] in the past. It has been implemented in systems with few components, but unfortunately, it becomes unfeasible for larger systems because the complexity grows exponentially with the system size. To solve this problem, another alternative technique called self-testing [9] was proposed. These two techniques could be used to verify the quantum systems.

Self-testing is a device-independent approach to verifying that the previously unknown quantum system state and uncharacterized measurement operators are to some degree close to the target state and measurements (up to local isometries) based only on observed statistics, without assuming the dimension of the quantum system. The device-independent (DI) approach [10] is important in practical quantum communications. One of the main applications of self-testing is quantum key distribution (QKD) [11,12], which is of great interest because of its high security. For the users, the quantum key distribution system is purchased from the device providers. However, if a device provider deliberately creates a “dishonest” quantum device, which does not perform key distribution according to the correct protocol, then the key distribution performed with such a device will be insecure. Therefore, it is imperative to test the trustworthiness of quantum cryptographic devices. Fortunately, based on the idea of self-testing quantum systems, it is possible to design device-independent quantum cryptography protocols. For example, in the device-independent QKD protocols, even if the device provider is not trusted, the user can still ensure that the keys generated by the device are secure. The essence is that the user self-tests the quantum device and uses its output as the key under the condition that the test is passed, and the key must be trusted in this case. In addition to quantum key distribution, various protocols, such as random number generation [13], and entanglement witness [14], have been designed in a device-independent framework so far.

Let us consider a scenario where *N* distant observers share an unknown *N*-partite quantum state |Ψ〉. Each party can perform uncharacterized measurements $\left\{M_{a_{i}}^{x_{i}}\right\}$ on the state with their quantum devices, where *i* marks different parties, $x_{i}$ marks different measurement settings for party *i*, and $a_{i}$ marks the corresponding measurement outcomes. In a device-independent scenario, the process of measuring an unknown quantum state can be viewed as a black box for the *N* observers: they can only query their devices with possible measurement settings $x_{i}$, and to any query, the black box produces a corresponding outcome. As we do not assume the dimension of the quantum system, the dimension of the Hilbert space is not fixed. Without loss of generality, we assume that the unknown state is pure. There is no loss of generality because an extra system can be added to some of the parties, if necessary, to purify the state, and the purification of the state can be included in the black boxes. Similarly, we can further assume that the measurement operators are projective without loss of generality, as an auxiliary system in some known state can be added to the measured system to replace a general POVM on this system by a projective measurement on the extended system [9]. According to the postulates of quantum mechanics [15], the data they observe are given by
(1)$$\begin{array}{c} p\left(a_{1},a_{2},\cdots ,a_{N}|x_{1},x_{2},\cdots ,x_{N}\right)=\langle \Psi |M_{a_{1}}^{x_{1}}⊗M_{a_{2}}^{x_{2}}⊗\cdots M_{a_{N}}^{x_{N}}|\Psi \rangle , \end{array}$$
which is referred to as a *correlation* [16] based on the quantum nonlocality [17] of entangled states [18]. As the possibility to self-test quantum states and measurements usually relies on quantum nonlocality, only the entangled states can be device-independently verified by self-testing techniques. The self-testing problem consists of deciding if the knowledge of the *correlation* allows us to deduce the structure of the unknown quantum system.

Symmetric states [19] have been found useful in many quantum information tasks, such as measurement-based quantum computation (MBQC) [20], as they are not too entangled to be computationally universal. Due to the important role of symmetry in the field of quantum entanglement, it is important to explore the properties of symmetric states.

This paper is organized as follows. The basic definitions and preliminaries are given in Section 2. In Section 3, we prove analytically that a particular symmetric four-qubit state can be self-tested and give bounds that are robust to inevitable experimental errors. In addition, we show the self-testing of a family of parameterized four-qubit symmetric states, which are superpositions of four-qubit Dicke states through projections onto two subsystems in Section 4, and we give the conclusions in Section 5.

## 2. Basic Definitions and Preliminaries

In this section, we present the definitions of self-testing [21] and give the known results as several lemmas, which may be used as building blocks for our work.

**Definition** **1**(Self-testing). *A known correlation allows for self-testing the state $|\Psi^{{}^{\prime }}\rangle$ and measurements $\left\{{M^{{}^{\prime }}}_{a_{i}}^{x_{i}}\right\}$; if any state and measurements |Ψ〉 and $\left\{M_{a_{i}}^{x_{i}}\right\}$ reproduce the correlation, there exists a local isometry* Φ *such that*
(2)$$\begin{array}{cc} \; & \Phi \left(|\Psi \rangle \right)=|junk\rangle ⊗|\Psi^{{}^{\prime }}\rangle ,\\ \Phi (M_{a_{1}}^{x_{1}}⊗M_{a_{2}}^{x_{2}}⊗\cdots ⊗M_{a_{N}}^{x_{N}} & |\Psi \rangle )=|junk\rangle ⊗\left({M^{{}^{\prime }}}_{a_{1}}^{x_{1}}⊗{M^{{}^{\prime }}}_{a_{2}}^{x_{2}}⊗\cdots ⊗{M^{{}^{\prime }}}_{a_{N}}^{x_{N}}|\Psi^{{}^{\prime }}\rangle \right), \end{array}$$
*where the state |junk〉 is an auxiliary state which will be traced out and thus not taken into consideration.*

The currently known self-testing protocols are mainly tailored for bipartite states [22,23,24,25,26]. We first review two-qubit self-testing. As given in [23,24], all pure two-qubit entangled states can be self-tested by observing the maximum violation of the tilted CHSH inequality [27]
(3)$$\begin{array}{c} B\left(\alpha ,A_{0},A_{1},B_{0},B_{1}\right)\equiv \alpha A_{0}+A_{0}\left(B_{0}+B_{1}\right)+A_{1}\left(B_{0}-B_{1}\right)\leq 2+\alpha , \end{array}$$
where 0≤α<2 and $A_{i}$ and $B_{i}$ are observables with outcomes ±1. The maximal violation is given by $b\left(\alpha \right)≜\max_{\phi }\langle \phi |B\left(\alpha ,A_{0},A_{1},B_{0},B_{1}\right)|\phi \rangle =\sqrt{8+2\alpha^{2}}$.

**Lemma** **1.**
*Any pure two-qubit states in their Schmidt form $|\Psi_{\theta }\rangle =\cos\theta |00\rangle +\sin\theta |11\rangle$ can be self-tested by achieving the maximal quantum violation of the tilted CHSH inequality Equation (Equation 3). The corresponding measurements $A_{i}$ and $B_{i}$ for two distant parties, Alice and Bob, are set as*

(4)
$$\begin{array}{c} A_{1}=\sigma_{z},B_{1}=\cos\mu \sigma_{z}+\sin\mu \sigma_{x},\\ A_{2}=\sigma_{x},B_{2}=\cos\mu \sigma_{z}-\sin\mu \sigma_{x}. \end{array}$$


*Here, $\sin2\theta =\sqrt{\frac{4-\alpha^{2}}{4+\alpha^{2}}}$ and μ=arctansin2θ.*


Especially for the maximally entangled two-qubit states in the form $\frac{|00\rangle +|11\rangle }{\sqrt{2}}$, there exist another two criteria [25].

**Lemma** **2**(Mayers–Yao criterion). *Consider five unknown dichotomic measurements $\left\{X_{A},Z_{A};X_{B},Z_{B},D_{B}\right\}$. If the following statistics are observed*
(5)$$\begin{array}{c} \langle \Psi |Z_{A}Z_{B}|\Psi \rangle =\langle \Psi |X_{A}X_{B}|\Psi \rangle =1,\\ \langle \Psi |X_{A}Z_{B}|\Psi \rangle =\langle \Psi |Z_{A}X_{B}|\Psi \rangle =0,\\ \langle \Psi |Z_{A}D_{B}|\Psi \rangle =\langle \Psi |X_{A}D_{B}|\Psi \rangle =\frac{1}{\sqrt{2}}, \end{array}$$
*then up to a local isometry, the state |Ψ〉 is self-tested into the maximally entangled two-qubit state $\frac{|00\rangle +|11\rangle }{\sqrt{2}}$, and the measurements are the suitable complementary Pauli operators.*

**Lemma** **3**(XOR game). *Consider four unknown operators $\left\{A_{0},A_{1},B_{0},B_{1}\right\}$ with binary outcomes ±1 and let $E_{xy}\equiv \langle \Psi |A_{x}B_{y}|\Psi \rangle =\cos\alpha_{xy}$. The state |Ψ〉 can be self-tested into the maximally entangled two-qubit state $\frac{|00\rangle +|11\rangle }{\sqrt{2}}$ by winning the binary nonlocal XOR game defined by the figure of merit $\sum_{\left(x,y\right)\in {\left(0,1\right)}^{2}}f_{xy}E_{xy}$ if it satisfies $\alpha_{00}+\alpha_{10}=\alpha_{01}-\alpha_{11}$. The coefficients $f_{xy}$ are constructed by*
(6)$$\left(\begin{array}{c} f_{00}\\ f_{01}\\ f_{10}\\ f_{11} \end{array}\right)=\left(\begin{array}{c} \sin^{-1}\alpha_{00}\\ -\sin^{-1}\left(\alpha_{00}+\alpha_{10}+\alpha_{11}\right)\\ \sin^{-1}\alpha_{10}\\ \sin^{-1}\alpha_{11} \end{array}\right).$$
*However, the self-testing of multipartite scenarios has not been fully explored. In this paper, we work on the four-qubit symmetric entangled states.*


**Definition** **2**(Symmetric states). *Symmetric quantum states preserve invariance under any permutation of their subsystems. We say that an n-partite state |Ψ〉 is symmetric if P|Ψ〉=|Ψ〉 for all $P\in S_{n}$, where $S_{n}$ is the symmetric group of n elements. The n-qubit Dicke states $|S_{n,k}\rangle$ are typical examples of symmetric state, which are the equally weighted sums of all permutations of computational basis states with n−k qubits being |0〉 and k being |1〉:*
(7)|Sn,k〉=nk−1/2∑Permutation|0〉|0〉⋯|0〉︸n−k|1〉|1〉⋯|1〉︸k.
*Let |Ψ〉 be a state vector in an N-fold tensor product space $S_{1}⊗\cdots ⊗S_{N}$, where $dimS_{1}=\cdots =dimS_{N}=d\geq 2$ and N≥3. As the generalization of the Schmidt decomposition given in [28], if d=2, any multipartite states can be written in the expansion as*

(8)
$$\begin{array}{c} |\Psi \rangle =\sum_{i_{1},i_{2},\cdots ,i_{N}\in \left\{0,1\right\}}t^{i_{1}i_{2}\cdots i_{N}}|i_{1}\rangle |i_{2}\rangle |i_{3}\rangle \cdots |i_{N}\rangle , \end{array}$$

*where some coefficients satisfy*

(9)
$$\begin{array}{c} t^{011\cdots 11}=t^{101\cdots 11}=\cdots =t^{111\cdots 10}=0, \end{array}$$

*and the rest $2^{N}-N$ orthogonal product states*

(10)
$$\begin{array}{c} \left\{|\overset{N}{\overset{︷}{000\cdots 00}}\rangle ,|\overset{N-1}{\overset{︷}{000\cdots 0}}1\rangle ,\cdots ,|1\overset{N-1}{\overset{︷}{0\cdots 00}}\rangle ,\cdots ,|00\overset{N-2}{\overset{︷}{1\cdots 11}}\rangle ,\cdots ,|\overset{N-2}{\overset{︷}{11\cdots 1}}00\rangle ,|\overset{N}{\overset{︷}{111\cdots 11}}\rangle \right\} \end{array}$$

*can be seen as a set of local bases. To characterize the symmetric multi-qubit states, we only need to make the rest coefficients have properties*

(11)
$$\begin{array}{cc} t^{000\cdots 01}=t^{00\cdots 010} & =\cdots =t^{100\cdots 00},\\ \; & \vdots \\ t^{001\cdots 11}=t^{0101\cdots 1} & =\cdots =t^{11\cdots 100}. \end{array}$$



## 3. Self-Testing of a Four-Qubit Symmetric State

In this section, we focus on a four-qubit symmetric state with a special structure by using the known results. In the case of N=4, as given in Equation (Equation 10), the set of local bases is
(12)$$\begin{array}{cc} \{ & |0000\rangle ,|0001\rangle ,|0010\rangle ,|0100\rangle ,|1000\rangle ,|0011\rangle ,\\ \; & |0101\rangle ,|0110\rangle ,|1001\rangle ,|1010\rangle ,|1100\rangle ,|1111\rangle \} \end{array}$$

### 3.1. Self-Testing of a Specific Four-Qubit Symmetric State

The specific four-qubit symmetric state we consider is
(13)$$\begin{array}{c} |\Psi_{1}^{{}^{\prime }}\rangle =\frac{1}{2\sqrt{2}}{\left(|0000\rangle +|0011\rangle +|0101\rangle +|0110\rangle +|1001\rangle +|1010\rangle +|1100\rangle +|1111\rangle \right)}_{ABCD}, \end{array}$$
which is shared by four distant observers, Alice, Bob, Charlie and David.

Rewrite the state as
(14)$$\begin{array}{cc} |\Psi_{1}^{{}^{\prime }}\rangle = & \frac{1}{2\sqrt{2}}[\sqrt{2}{|00\rangle }_{AB}⊗\frac{1}{\sqrt{2}}{\left(|00\rangle +|11\rangle \right)}_{CD}+\sqrt{2}{|01\rangle }_{AB}⊗\frac{1}{\sqrt{2}}{\left(|01\rangle +|10\rangle \right)}_{CD}\\ \; & +\sqrt{2}{|10\rangle }_{AB}⊗\frac{1}{\sqrt{2}}{\left(|01\rangle +|10\rangle \right)}_{CD}+\sqrt{2}{|11\rangle }_{AB}⊗\frac{1}{\sqrt{2}}{\left(|00\rangle +|11\rangle \right)}_{CD}]. \end{array}$$

The concept of partial measurements [29] is involved in our scheme, which appears very often in reality. A similar approach for quantum nonlocality chracterization is given in [30], where quantum imcompatibility is used to characterize nonlocality. According to the partial measurement postulate given in [29], if any two parties, without loss of generality, e.g., Alice and Bob, each measure in the $\sigma_{z}$ basis, the remaining two parties share a maximally entangled two-qubit state $\frac{|00\rangle +|11\rangle }{\sqrt{2}}$ conditioned on the outcome “00” and “11”, respectively, which can be self-tested combining Lemma 2.

We construct the local isometry Φ as Figure 1. Here, *H* is the usual Hadamard gate. Obviously, if $Z_{i}=\sigma_{z},X_{i}=\sigma_{x}$, we can extract the essential information on the unknown state into auxiliary systems. Inspired by this, $Z_{i}$ and $X_{i}$ should act analogously to the Pauli operators on $|\Psi_{1}\rangle$ to guarantee the feasibility of the protocol. However, in order to make the protocol device-independent, we cannot directly consider $Z_{i}$ and $X_{i}$ of each party as Pauli operators, but should construct them with the measurements $\left\{M_{a_{i}}^{x_{i}}\right\}$ properly. We sum the result up as below.

**Result** **1.**
*Consider four spatially separated parties, Alice, Bob, Charlie and David, each performing three measurements $\left\{X_{s},Z_{s},M_{s}\right\}\left(s\in \left\{A,B,C,D\right\}\right)$ with binary outcomes on an unknown shared quantum state $|\Psi_{1}\rangle$. The target symmetric state $|\Psi_{1}^{{}^{\prime }}\rangle$ is self-tested if the statistics are observed as the following:*

(15)
$$\begin{array}{cc} \; & \langle P_{A}^{0}P_{B}^{0}P_{C}^{0}P_{D}^{0}\rangle =\langle P_{A}^{0}P_{B}^{0}P_{C}^{1}P_{D}^{1}\rangle =\langle P_{A}^{0}P_{B}^{1}P_{C}^{0}P_{D}^{1}\rangle =\langle P_{A}^{0}P_{B}^{1}P_{C}^{1}P_{D}^{0}\rangle \\ \; & =\langle P_{A}^{1}P_{B}^{0}P_{C}^{0}P_{D}^{1}\rangle =\langle P_{A}^{1}P_{B}^{0}P_{C}^{1}P_{D}^{0}\rangle =\langle P_{A}^{1}P_{B}^{1}P_{C}^{0}P_{D}^{0}\rangle =\langle P_{A}^{1}P_{B}^{1}P_{C}^{1}P_{D}^{1}\rangle =\frac{1}{8}, \end{array}$$


(16)
$$\left\{\begin{array}{c} \langle P_{i}^{0}P_{j}^{0}X_{k}X_{l}\rangle =\langle P_{i}^{0}P_{j}^{0}Z_{k}Z_{l}\rangle =\frac{1}{4}\\ \langle P_{i}^{0}P_{j}^{0}X_{k}M_{l}\rangle =\langle P_{i}^{0}P_{j}^{0}Z_{k}M_{l}\rangle =\frac{1}{4\sqrt{2}}\\ \langle P_{i}^{0}P_{j}^{0}X_{k}Z_{l}\rangle =0 \end{array}\right.,\left\{\begin{array}{c} \langle P_{A}^{1}P_{B}^{1}X_{C}X_{D}\rangle =\langle P_{A}^{1}P_{B}^{1}Z_{C}Z_{D}\rangle =\frac{1}{4}\\ \langle P_{A}^{1}P_{B}^{1}X_{C}M_{D}\rangle =\langle P_{A}^{1}P_{B}^{1}Z_{C}M_{D}\rangle =\frac{1}{4\sqrt{2}}\\ \langle P_{i}^{1}P_{j}^{1}X_{k}Z_{l}\rangle =0 \end{array}\right.,$$

*where (i,j,k,l)={(A,B,C,D),(A,C,B,D),(A,D,B,C),(B,C,A,D),(B,D,A,C),(C,D,A,B)} and $P_{s}^{0}≜P_{Z_{s}=+1}=\frac{1+Z_{s}}{2},P_{s}^{1}≜P_{Z_{s}=-1}=\frac{1-Z_{s}}{2}$, where s∈{A,B,C,D} are projectors for the $Z_{s}$ measurement.*


**Proof.** To begin with, the output after the isometry given in Figure 1 is
(17)$$\begin{array}{cc} |{\tilde{\Psi }}_{1}\rangle & =\Phi (|\Psi_{1}\rangle {|0000\rangle }_{A^{{}^{\prime }}B^{{}^{\prime }}C^{{}^{\prime }}D^{{}^{\prime }}})\\ \; & =\sum_{a,b,c,d\in \{0,1\}}X_{A}^{a}X_{B}^{b}X_{C}^{c}X_{D}^{d}P_{A}^{a}P_{B}^{b}P_{C}^{c}P_{D}^{d}|\Psi_{1}\rangle |abcd\rangle . \end{array}$$Observation Equation (Equation 15) implies that
(18)$$\begin{array}{cc} \; & \langle P_{A}^{0}P_{B}^{0}P_{C}^{0}P_{D}^{0}\rangle +\langle P_{A}^{0}P_{B}^{0}P_{C}^{1}P_{D}^{1}\rangle +\langle P_{A}^{0}P_{B}^{1}P_{C}^{0}P_{D}^{1}\rangle +\langle P_{A}^{0}P_{B}^{1}P_{C}^{0}P_{D}^{1}\rangle \\ \; & +\langle P_{A}^{1}P_{B}^{0}P_{C}^{0}P_{D}^{1}\rangle +\langle P_{A}^{1}P_{B}^{0}P_{C}^{1}P_{D}^{0}\rangle +\langle P_{A}^{1}P_{B}^{1}P_{C}^{0}P_{D}^{0}\rangle +\langle P_{A}^{1}P_{B}^{1}P_{C}^{1}P_{D}^{1}\rangle =1, \end{array}$$
and thus $P_{A}^{a}P_{B}^{b}P_{C}^{c}P_{D}^{d}|\Psi_{1}\rangle =0$ for other eight projectors. Based on the fact that 〈ψ|ϕ〉=1 implies |ψ〉=|ϕ〉, observation of Equation (Equation 16) implies
(19)$$\left\{\begin{array}{c} P_{i}^{0}P_{j}^{0}X_{k}|\Psi_{1}\rangle =P_{i}^{0}P_{j}^{0}X_{l}|\Psi_{1}\rangle ,P_{i}^{0}P_{j}^{0}Z_{k}|\Psi_{1}\rangle =P_{i}^{0}P_{j}^{0}Z_{l}|\Psi_{1}\rangle \\ P_{i}^{0}P_{j}^{0}X_{k}|\Psi_{1}\rangle ⊥P_{i}^{0}P_{j}^{0}Z_{k}|\Psi_{1}\rangle ,P_{i}^{0}P_{j}^{0}X_{l}|\Psi_{1}\rangle ⊥P_{i}^{0}P_{j}^{0}Z_{l}|\Psi_{1}\rangle \\ P_{i}^{0}P_{j}^{0}M_{l}|\Psi_{1}\rangle =\frac{P_{i}^{0}P_{j}^{0}X_{l}|\Psi_{1}\rangle +P_{i}^{0}P_{j}^{0}Z_{l}|\Psi_{1}\rangle }{\sqrt{2}}=\frac{P_{i}^{0}P_{j}^{0}X_{k}|\Psi_{1}\rangle +P_{i}^{0}P_{j}^{0}Z_{k}|\Psi_{1}\rangle }{\sqrt{2}} \end{array}\right.,$$
and
(20)$$\left\{\begin{array}{c} P_{A}^{1}P_{B}^{1}X_{C}|\Psi_{1}\rangle =P_{A}^{1}P_{B}^{1}X_{D}|\Psi_{1}\rangle ,P_{A}^{1}P_{B}^{1}Z_{C}|\Psi_{1}\rangle =P_{A}^{1}P_{B}^{1}Z_{D}|\Psi_{1}\rangle \\ P_{A}^{1}P_{B}^{1}X_{C}|\Psi_{1}\rangle ⊥P_{A}^{1}P_{B}^{1}Z_{C}|\Psi_{1}\rangle ,P_{A}^{1}P_{B}^{1}X_{D}|\Psi_{1}\rangle ⊥P_{A}^{1}P_{B}^{1}Z_{D}|\Psi_{1}\rangle \\ P_{A}^{1}P_{B}^{1}M_{D}|\Psi_{1}\rangle =\frac{P_{A}^{1}P_{B}^{1}X_{D}|\Psi_{1}\rangle +P_{A}^{1}P_{B}^{1}Z_{D}|\Psi_{1}\rangle }{\sqrt{2}}=\frac{P_{A}^{1}P_{B}^{1}X_{C}|\Psi_{1}\rangle +P_{A}^{1}P_{B}^{1}Z_{C}|\Psi_{1}\rangle }{\sqrt{2}} \end{array}\right..$$Obviously, we have ${\left(P_{i}^{0}P_{j}^{0}M_{l}\right)}^{2}|\Psi_{1}\rangle =P_{i}^{0}P_{j}^{0}M_{l}^{2}|\Psi_{1}\rangle$. Since $X^{2}=Z^{2}=M^{2}=I$, we have $P_{i}^{0}P_{j}^{0}|\Psi_{1}\rangle =\frac{P_{i}^{0}P_{j}^{0}{\left(X_{k}+Z_{k}\right)}^{2}|\Psi_{1}\rangle }{2}=\frac{P_{i}^{0}P_{j}^{0}{\left(X_{l}+Z_{l}\right)}^{2}|\Psi_{1}\rangle }{2}$. Hence, we obtain the following anti-commutation relation
(21)$$\begin{array}{cc} \; & P_{i}^{0}P_{j}^{0}X_{k}Z_{k}|\Psi_{1}\rangle =-P_{i}^{0}P_{j}^{0}Z_{k}X_{k}|\Psi_{1}\rangle \\ \; & P_{i}^{0}P_{j}^{0}X_{l}Z_{l}|\Psi_{1}\rangle =-P_{i}^{0}P_{j}^{0}Z_{l}X_{l}|\Psi_{1}\rangle \end{array}$$
for all (i,j,k,l)={(A,B,C,D),(A,C,B,D),(A,D,B,C),(B,C,A,D),(B,D,A,C),(C,D,A,B)}, and similarly,
(22)$$\begin{array}{cc} \; & P_{A}^{1}P_{B}^{1}X_{C}Z_{C}|\Psi_{1}\rangle =-P_{A}^{1}P_{B}^{1}Z_{C}X_{C}|\Psi_{1}\rangle \\ \; & P_{A}^{1}P_{B}^{1}X_{D}Z_{D}|\Psi_{1}\rangle =-P_{A}^{1}P_{B}^{1}Z_{D}X_{D}|\Psi_{1}\rangle . \end{array}$$All these properties of the operators will help to reduce the output Equation (Equation 17). By using Equation (Equation 21), $X_{C}X_{D}P_{A}^{0}P_{B}^{0}P_{C}^{1}P_{D}^{1}|\Psi_{1}\rangle$ is equal to $P_{A}^{0}P_{B}^{0}P_{C}^{0}X_{C}P_{D}^{0}X_{D}|\Psi_{1}\rangle$. As $P_{A}^{0}P_{B}^{0}X_{C}|\Psi_{1}\rangle =P_{A}^{0}P_{B}^{0}X_{D}|\Psi_{1}\rangle$ shown in Equation (Equation 19), this term becomes $P_{A}^{0}P_{B}^{0}P_{C}^{0}P_{D}^{0}|\Psi_{1}\rangle$. We can simplify the other five terms similarly. For the last term, we can obtain $P_{A}^{1}P_{B}^{1}P_{C}^{0}P_{D}^{0}|\Psi_{1}\rangle$ using Equations (Equation 20) and (Equation 22), which can also be simplified to $P_{A}^{0}P_{B}^{0}P_{C}^{0}P_{D}^{0}|\Psi_{1}\rangle$. As a reminder, there are eight terms equal to zero. Hence, the output Equation (Equation 17) is reduced to
(23)$$\begin{array}{cc} |\Psi_{1}^{*}\rangle = & P_{A}^{0}P_{B}^{0}P_{C}^{0}P_{D}^{0}|\Psi_{1}\rangle (|0000\rangle +|0011\rangle +|0101\rangle +|0110\rangle +|1001\rangle +|1010\rangle +|1100\rangle \\ \; & +|1111\rangle ) \end{array}$$
and can be normalized into the form of $|junk\rangle ⊗|\Psi_{1}^{{}^{\prime }}\rangle$, here $|junk\rangle =2\sqrt{2}P_{A}^{0}P_{B}^{0}P_{C}^{0}P_{D}^{0}|\Psi_{1}\rangle$. □

### 3.2. Robustness Analysis Based on the $L_{2}$ Norm

In this section, we give the analysis of robustness based on the vector norm inequality. Result 1 relies on the observation of Equations (Equation 15) and (Equation 16) exactly; however, which may be impossible in actual experiments due to the inevitable deviation from the ideal case. Suppose each observation in Equations (Equation 15) and (Equation 16) admits a deviation at most ϵ around the ideal value. We say that the self-testing of $|\Psi_{1}^{{}^{\prime }}\rangle$ is robust [31] if the isometry still extracts a state close to it and satisfies
(24)$$\begin{array}{c} ∥|{\tilde{\Psi }}_{1}\rangle -|junk\rangle ⊗|\Psi_{1}^{{}^{\prime }}\rangle ∥\leq f(\epsilon ), \end{array}$$
where f(ϵ)→0 when ϵ→0.

We show that
(25)$$\begin{array}{c} ∥|{\tilde{\Psi }}_{1}\rangle -|junk\rangle ⊗|\Psi_{1}^{{}^{\prime }}\rangle ∥\leq f\left(\epsilon \right)=265.98\epsilon +348.45\epsilon^{\frac{3}{4}}+94.87\epsilon^{\frac{1}{2}}+60.70\epsilon^{\frac{1}{4}} \end{array}$$
in Appendix A, which proves the robustness of Result 1.

## 4. Self-Testing of a Family of Parameterized Four-Qubit Symmetric States

In this part, we consider a more general state
(26)$$\begin{array}{cc} |\Psi_{2}^{{}^{\prime }}\rangle = & \frac{1}{\sqrt{8+4t^{2}}}[|0000\rangle +t\left(|0001\rangle +|0010\rangle +|0100\rangle +|1000\rangle \right)+|0011\rangle \\ \; & {+|0101\rangle +|0110\rangle +|1001\rangle +|1010\rangle +|1100\rangle +|1111\rangle ]}_{ABCD} \end{array}$$
where t>0 and t≠1. The parameterized state is a superposition of *W* state, GHZ state and $|S_{4,2}\rangle$ state, where the ratio of the coefficient of GHZ state and $|S_{4,2}\rangle$ state is a constant value, which is equal to $\frac{1}{\sqrt{3}}$. Rewrite the states as
(27)$$\begin{array}{cc} |\Psi_{2}^{{}^{\prime }}\rangle = & \frac{1}{\sqrt{8+4t^{2}}}[\sqrt{2+2t^{2}}{|00\rangle }_{AB}⊗\frac{1}{\sqrt{2+2t^{2}}}{\left(|00\rangle +t|01\rangle +t|10\rangle +|11\rangle \right)}_{CD}\\ \; & +\sqrt{2+t^{2}}{|01\rangle }_{AB}⊗\frac{1}{\sqrt{2+t^{2}}}{\left(t|00\rangle +|01\rangle +|10\rangle \right)}_{CD}\\ \; & +\sqrt{2+t^{2}}{|10\rangle }_{AB}⊗\frac{1}{\sqrt{2+t^{2}}}{\left(t|00\rangle +|01\rangle +|10\rangle \right)}_{CD}\\ \; & +\sqrt{2}{|11\rangle }_{AB}⊗\frac{1}{\sqrt{2}}{\left(|00\rangle +|11\rangle \right)}_{CD}]. \end{array}$$

Denote
(28)$$\begin{array}{cc} \; & |\psi_{1}\rangle =\frac{1}{\sqrt{2+2t^{2}}}{\left(|00\rangle +t|01\rangle +t|10\rangle +|11\rangle \right)}_{CD},\\ \; & |\psi_{2}\rangle =\frac{1}{\sqrt{2}}{\left(|00\rangle +|11\rangle \right)}_{CD}. \end{array}$$

The state $|\psi_{1}\rangle$ in its Schmidt form is
(29)$$\begin{array}{cc} \; & |\psi_{1}\rangle =\cos\beta {|0^{{}^{\prime }}\rangle }_{C}{|0^{{}^{\prime }}\rangle }_{D}+\sin\beta {|1^{{}^{\prime }}\rangle }_{C}{|1^{{}^{\prime }}\rangle }_{D}, \end{array}$$
where $\cos\beta =\frac{1+t}{\sqrt{2+2t^{2}}},\sin\beta =\frac{\left|1-t\right|}{\sqrt{2+2t^{2}}}$. Here, $\left\{{|i^{{}^{\prime }}\rangle }_{C}\right\}$, $\left\{{|i^{{}^{\prime }}\rangle }_{D}\right\},i\in \left\{0,1\right\}$ are the corresponding new bases for *C* and *D*. (See detail in Appendix C).

If t=1, $|\psi_{1}\rangle$ is not an entangled state and the lack of nonlocality may result in the failure of the self-testing. Following the framework of [32], we intend to divide the four parties into two parts, and one of them performs local measurements on $|\Psi_{2}\rangle$. If we divide ABCD randomly into groups that each have two parties, for example, AB and CD, as a result, the projection measurements may collapse the state shared by the remaining parts into some unknown pure bipartite entangled states. Then the remaining two parts should check whether the projected state they share violates maximally Equation (Equation 3) for the appropriate α. Without loss of generality, if A and B perform the measurement in the $\sigma_{z}$ bases, $|\psi_{1}\rangle$ and $|\psi_{2}\rangle$ should be self-tested by C and D, respectively, and simultaneously conditioned on the outcomes “00” and “11”.

Following the result given in Lemma 1, $|\psi_{1}\rangle$ can be self-tested by reaching the maximal violation of the tilted CHSH Bell inequality
(30)$$\begin{array}{c} b\left(\alpha \right)=\sqrt{8+2\alpha^{2}}=\frac{2\sqrt{2}\left(t+1\right)}{\sqrt{1+t^{2}}}, \end{array}$$
where $\alpha =2\sqrt{\frac{1-\sin^{2}2\beta }{1+\sin^{2}2\beta }}$ and the optimal measurement are set as Lemma 1 with $\tan\mu =\frac{1-t^{2}}{1+t^{2}}$. Meanwhile, $|\psi_{2}\rangle$ is still a maximally entangled two-qubit state under the same transformation of bases
(31)$$\begin{array}{c} |\psi_{2}\rangle =\frac{1}{\sqrt{2}}{\left(|0^{{}^{\prime }}0^{{}^{\prime }}\rangle +|1^{{}^{\prime }}1^{{}^{\prime }}\rangle \right)}_{CD}, \end{array}$$
and hence, we can use the same measurement settings as $|\psi_{1}\rangle$. As the definition given in Lemma 3, $\alpha_{00}=\mu ,\alpha_{01}=-\mu ,\alpha_{10}=\frac{\pi }{2}-\mu ,\alpha_{11}=-\frac{\pi }{2}-\mu$, and thus it will satisfy the condition $\alpha_{00}+\alpha_{10}=\alpha_{01}-\alpha_{11}$.

Define
(32)$$\begin{array}{c} f\left(t\right)=\left\{\begin{array}{c} 0,\;t<1\\ 1,\;t>1 \end{array}\right.. \end{array}$$

Then $|\psi_{1}\rangle$ can be self-tested by winning the XOR game and we give the criterion to self-test $|\Psi_{2}^{{}^{\prime }}\rangle$ as the following Result 2.

**Result** **2**(See proof in Appendix B). *Consider four spatially separated parties, Alice, Bob, Charlie and David, each performing five measurements with binary outcomes denoted as $A_{i},B_{j},C_{k},D_{l}\left(i,j,k,l\in \left\{0,1,2,3,4\right\}\right)$ on an unknown shared quantum state $|\Psi_{2}\rangle$. The target state $|\Psi_{2}^{{}^{\prime }}\rangle$ is self-tested if the statistics are observed as the following*
(33)$$\left\{\begin{array}{cc} \langle P_{A}^{0}P_{B}^{0}P_{C}^{0}P_{D}^{0}\rangle & =\langle P_{A}^{0}P_{B}^{0}P_{C}^{1}P_{D}^{1}\rangle =\langle P_{A}^{0}P_{B}^{1}P_{C}^{0}P_{D}^{1}\rangle =\langle P_{A}^{0}P_{B}^{1}P_{C}^{1}P_{D}^{0}\rangle =\langle P_{A}^{1}P_{B}^{0}P_{C}^{0}P_{D}^{1}\rangle \\ \; & =\langle P_{A}^{1}P_{B}^{0}P_{C}^{1}P_{D}^{0}\rangle =\langle P_{A}^{1}P_{B}^{1}P_{C}^{0}P_{D}^{0}\rangle =\langle P_{A}^{1}P_{B}^{1}P_{C}^{1}P_{D}^{1}\rangle =\frac{1}{8+4t^{2}}\\ \langle P_{A}^{0}P_{B}^{0}P_{C}^{0}P_{D}^{1}\rangle & =\langle P_{A}^{0}P_{B}^{0}P_{C}^{1}P_{D}^{0}\rangle =\langle P_{A}^{0}P_{B}^{1}P_{C}^{0}P_{D}^{0}\rangle =\langle P_{A}^{1}P_{B}^{0}P_{C}^{0}P_{D}^{0}\rangle =\frac{t^{2}}{8+4t^{2}} \end{array}\right.,$$
(34)$$\left\{\begin{array}{c} \langle P_{M}^{0}P_{N}^{0}B\left(\alpha ,Q_{0},Q_{1},R_{2},R_{3}\right)\rangle =\frac{t^{2}+1}{2t^{2}+4}\sqrt{8+2\alpha^{2}}\\ \langle P_{M}^{0}P_{N}^{0}Q_{0}\left(R_{2}-R_{3}\right)\rangle =0\\ \frac{\langle P_{M}^{0}P_{N}^{0}R_{2}\rangle }{2\sin\mu }-\frac{\langle P_{M}^{0}P_{N}^{0}R_{3}\rangle }{2\sin\mu }={\left(-1\right)}^{f(t)}\langle P_{M}^{0}P_{N}^{0}R_{1}\rangle \end{array}\right.,$$
(35)$$\left\{\begin{array}{c} \sum_{i\in (0,1),j\in (2,3)}f_{ij}\langle P_{A}^{1}P_{B}^{1}C_{i}Q_{j}\rangle =\frac{2}{(2+t^{2})\sin2\mu }\\ \langle P_{A}^{1}P_{B}^{1}C_{0}\left(D_{2}-D_{3}\right)\rangle =0\\ \frac{\langle P_{A}^{1}P_{B}^{1}D_{2}\rangle }{2\sin\mu }-\frac{\langle P_{A}^{1}P_{B}^{1}D_{3}\rangle }{2\sin\mu }={\left(-1\right)}^{f(t)}\langle P_{A}^{1}P_{B}^{1}D_{1}\rangle \end{array}\right.,$$
(36)$$\begin{array}{c} \langle P_{M}^{0}P_{N}^{0}P_{Q}^{0}P_{R}^{0}R_{0}\rangle =\frac{t}{4t^{2}+8}, \end{array}$$
*where (M,N,Q,R)∈{(A,B,C,D),(A,C,B,D),(A,D,B,C),(B,C,D,A),(B,D,A,C),(C,D,A,B)}, $P_{s}^{0}≜P_{Z_{s}=+1}=\frac{1+Z_{s}}{2},P_{s}^{1}≜P_{Z_{s}=-1}=\frac{1-Z_{s}}{2}$, where s∈{A,B,C,D} are projectors for the $Z_{s}$ measurement and $\sin\mu =\frac{\left|1-t\right|}{\sqrt{2+2t^{2}}},\cos\mu =\frac{1+t}{\sqrt{2+2t^{2}}}$. At the same time, we find a proper construction of the local isometry Φ, where $Z_{s}$ and $X_{s}$ are based on the measurement settings*
(37)$$\begin{array}{ccc} \; & Z_{A}=A_{1}={\left(-1\right)}^{f(t)}\frac{A_{2}-A_{3}}{2\sin\mu }, & X_{A}=A_{0}=\frac{A_{2}+A_{3}}{2\cos\mu },\\ \; & Z_{B}=B_{1}={\left(-1\right)}^{f(t)}\frac{B_{2}-B_{3}}{2\sin\mu }, & X_{B}=B_{0}=\frac{B_{2}+B_{3}}{2\cos\mu },\\ \; & Z_{C}=C_{1}={\left(-1\right)}^{f(t)}\frac{C_{2}-C_{3}}{2\sin\mu }, & X_{C}=C_{0}=\frac{C_{2}+C_{3}}{2\cos\mu },\\ \; & Z_{D}=D_{1}={\left(-1\right)}^{f(t)}\frac{D_{2}-D_{3}}{2\sin\mu }, & X_{D}=D_{0}=\frac{D_{2}+D_{3}}{2\cos\mu }, \end{array}$$
*and thus makes the protocol device-independent. In addition, each party may need another fifth measurements $A_{4}=Z_{A}X_{A},B_{4}=Z_{B}X_{B},C_{4}=Z_{C}X_{C},D_{4}=Z_{D}X_{D}$ to obtain the observation of Equation (Equation 36). Since $\sigma_{Z}\sigma_{X}=i\sigma_{Y}$, the fifth measurements are feasible in practical experiments.*

## 5. Conclusions

In this paper, we propose schemes to self-test a large family of four-qubit symmetric states. The target states we focus on are the superposition of the four-qubit Dicke states.

We first present a procedure for self-testing of a particular four-qubit symmetric state with a special structure, and this procedure makes use of the self-testing of the maximally entangled two-qubit state $\frac{|00\rangle +|11\rangle }{\sqrt{2}}$. At the same time, we prove that this protocol is robust against inevitable experimental errors based on norm inequality. In addition, we propose an approach to self-test a one-parameter family of four-qubit pure states through projections onto two systems. Here in our work, only the simplest Pauli measurements are used, which is quite helpful in the experiments.

It would also be of interest to work on a more general state with two parameters by using the swap method and semidefinite programming (SDP) [26] in the form
(38)$$|\Psi \rangle =\cos\theta \cos\rho |GHZ\rangle +\cos\theta \sin\rho |S_{4,2}\rangle +\sin\theta |W\rangle ,$$
where $\theta \in \left[0,\frac{\pi }{2}\right],\rho \in \left[0,\frac{\pi }{2}\right]$, which may provide better robustness than the analytical bounds. What is more, our work could potentially be generalized to a higher dimension scenario. These are reserved for our future work.

## Figures and Tables

**Figure 1 entropy-24-01003-f001:**
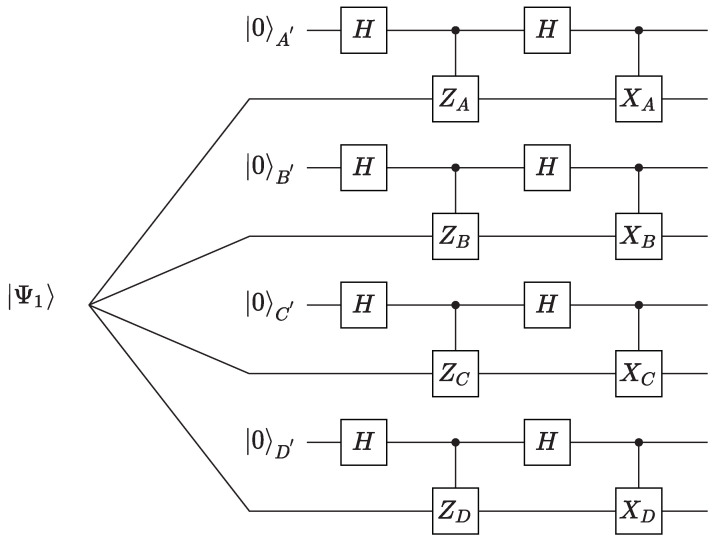
Swap circuit of the isometry Φ to self-test the target state $|\Psi_{1}^{{}^{\prime }}\rangle$.

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
