# Peer review of "Robust Self-Testing of Four-Qubit Symmetric States"

_entropy, 2022, doi:10.3390/e24071003_

Round 1
Reviewer 1 Report
This article offers a method of self-testing, i.e. verification of accuracy of output, for quantum devices. An alternative to quantum state tomography is proposed: a method of self-testing that is a device-independent approach. This method is presented in a clear and understandable manner and is applied to a family of four-qubit symmetric states. The authors demonstrate that their approach is robust against experimental errors based on norm inequality. More general self-tests for larger systems are suggested. The article is educational and should interest researchers in quantum systems. I support the article for publication in its present form.
Reviewer 2 Report
This is a report for the manuscript titled ``Robust Self-testing of Four-qubit Symmetric States''. In this manuscript, the authors present a self-testing protocol for verifying a family of symmetric four-qubit states.
They present self-testing as a device-independent approach to verify an unknown quantum state. They present this as an alternative technique to quantum state tomography, which requires a prohibitively expensive number of measurements. The motivation to find an alternative to state tomography is justifiable. While previous works have focused on self-testing of symmetric three-qubit states, the authors claim that they provide a self-testing protocol for four-qubit states for the first time. The results presented by the authors are rigorous, and explained well.
However, I cannot recommend this article for publication in a peer-reviewed journal. In my opinion, the results presented here are too simple to justify publication in a journal. The paper's main results can be explained simply as follows: One can consider the special symmetric state defined in Eq.(14), work out all the $4^4 = 64$ possible measurement outcomes on paper or computer, realize that the state has several symmetries -- permutational symmetries of qubits and rotation symmetries of measurement axes, and therefore only a smaller set of outcomes matter. One can then easily arrive at the conclusions in Eqs.(15) and (16). I admit this is an overly simplified summary, but it is not far from the true summary. A similar argument can be formulated for the family of states in Eq.(26).
Reviewer 3 Report
Dear authors,
You studied and analyzed the Four symmetric states to prove the Robust sel-testing of this system. It is an interesting research work. There are some questions for you, please reply to them: 1. As you mentioned, the self-testing is a quantum state verification technology, so I would like to know what you want to verify, some metrics should be mentioned, for instance how much entangles the state is, maybe? 2. In the fourth paragraph you say that "As we do not assume the dimension, the state can be considered as pure and the measurements are projective." I can rarely understand if that is a enough reason to considerar th pure state only. Please explain the reason. 3. Line 27, "which is referred to as a correlation [12] based on the quantum nonlocality [13] of entangled states" , so you will only consider the entangled state? Please make it clear. 4. The condition of the eq.(3) where alpha >=0, is not concurrence with the condition of eq.(4) sin2theta = sqrt(4- alpha2)/---. 5. In eq.(5) the first equation, the second part, if you use the singlet state, the inner product is -1, not 1. 6. What is the meaning of "d" before eq. (8)? 7. In the eq.(16), how to design the M operator? 8. In the eq. (26), you say "where the ratio of the coefficient of GHZ state and |S4,2⟩ state is a constant value equals to 1 / sqrt(3), why? WaAuthor Response
Please see the attachment.

Reviewer 4 Report
In this work, the authors extended the results of self-testing from 2-qubits and 3-qubits ststes to four qubits state. They first give a criterion for self-testing of a four-qubit symmetric state with a special structure and the robustness analysis based on vector norm inequalities. Then they generalized to a family of parameterized four-qubit symmetric states through projections onto two subsystems. It is useful contribution to the foundations of quantum information. I recommend its acceptance after minor revision.
In the revision, it is better to point out that quantum self-testing is closely related to quantum state tomography. Some related points need attention. Normally, QKD is regarded as quantum communication as is reviewed in ref. [2]. In the strict sense, QKD is key negotiation rather than communication. In quantum version, quantum secure direct communication is quantum communication [r1]. A recent review on chip-based QKD can be found in ref. [r2]. In quantum self-testing, measurement is used. A generalized measurement is given in ref. [r3], where the outcome of a partial measurement is given. Measurement on part of a quantum system is often met issue in quantum information. A similar approach for quantum nonlocality chracterization is given ref. [r4],where quantum imcompatibility is used to characterize nonlocality.
[r1] Long, Gui-Lu, and Xiao-Shu Liu. "Theoretically efficient high-capacity quantum-key-distribution scheme." Physical Review A 65.3 (2002): 032302.
[r2] Kwek, Leong-Chuan, et al. "Chip-based quantum key distribution." AAPPS Bulletin 31.1 (2021): 15
[r3] Long, GuiLu. "Collapse-in and collapse-out in partial measurement in quantum mechanics and its wise interpretation." Science China Physics, Mechanics, and Astronomy 64.8 (2021): 280321.
[r4] Zhang, Xiaolin, et al. "A geometrical framework for quantum incompatibility resources." AAPPS Bulletin 32.1 (2022): 17
Reviewer 5 Report
This paper addresses the problem of reconstructing a multi-particle state on the basis of the statistics it gives rise to. In particular, the authors consider entangled states of four qubits, shared by four distant observers who perform local measurements and find non-classical correlations that maximally violate Bell-type inequalities. The results presented strike me as correct, and as potentially helpful in experimental practice (in situations in which it should be verified which state has been prepared). The text is a bit sloppy in some places, and a grammatical check would enhance the paper's readability.
Round 2
Reviewer 2 Report
This is my second report as Reviewer 2.
The authors have responded to my comments by explaining the significance of self-testing and explaining why self-testing is not trivial. I thank the authors for their explanation. I thank them for clarifying that the benefit of self-testing is that it is device-independent.
However, I still have reservations about the simplicity/complexity of the idea. Let us suppose we want to self-test the singlet state, $(\ket{01} - \ket{10})/\sqrt{2}$. The simplest way in which the singlet state is taught to a new student is by showing the following properties: $\braket{ X_1 X_2 } = -1$, $\braket{ X_1 Z_2 } = 0$, and full SU(2) symmetry of the state, where X, Y, Z are Pauli operators. Further, the student is also taught that $X$, $Y$ and $Z$ can be arbitrary combinations of Pauli operators while satisfying the right commutation relations, due to the SU(2) symmetry. This is all standard course material for an undergraduate (or graduate) quantum mechanics course. The (first two lines of) Mayers-Yao criterion, which the authors kindly explained to me, is exactly the same as what I've written above, for unknown operators $A_0, A_1, B_0, B_1, B_2$. The authors' extension of the Mayers-Yao criterion to four-qubit states (Result 1) is trivial, because upon measuring the first two qubits, the state collapses to a state that is isometric to the singlet. My view of \textit{this} paper is that it does not do anything more than the level of a quantum mechanics course homework problem. Therefore, I do not recommend it for publication.
Reviewer 3 Report
The comment 5: Please check what is a singlet state. I was so surprised by this mistake again. That is a very basic concept in quantum mechanics, it can not be wrongly used. Even though you cited the Mayers-Yao criterion, remember that not any published work is correctly done. And please check all of the consequence.
Round 3
Reviewer 2 Report
This is my third report. In their response to my second report, the authors have not satisfactorily addressed my concern that the whole point of their paper is trivial.
The authors claim that the whole point of their article is designing the local isometry transform. However, the local isometry transform (Figure 1 in this manuscript) is identical to the local isometry transform in, for example, PRA 90, 042339 (which is cited as reference 23 in their manuscript), except for the number of the qubits in the respective circuits. According to the authors’ own admission, the whole point of their article is already published elsewhere, thus voiding their claim to publishing this manuscript.
Moreover, I do not understand why the authors write “the local isometry is derived by some known special properties of quantum mechanics, rather than conceptual speculation”. Who is speculating what here? It seems completely irrelevant to my report, and completely irrelevant to the preceding and succeeding sentences in the authors’ response letter.
The authors mention that McKague et al and Wu et al have used a similar method for self-testing of 3-qubit states. I agree with this statement, but I also re-emphasize that my report is based on the contents of this manuscript only.
In summary, the authors have not convinced me that they have accomplished anything non-trivial. In fact, they have admitted that the whole point of their article is already published in the literature. I respectfully stay with my decision to not publish this paper.
Reviewer 3 Report
Congratulations!
